# Relationship Between Clinical Factors and the Risk of Cerebral Vasospasm Following Aneurysmal Subarachnoid Hemorrhage: An Exploratory Analysis

**DOI:** 10.3390/life16010059

**Published:** 2025-12-30

**Authors:** Máté Czabajszki, Attila Garami, Péter Csécsei, Béla Viskolcz, Csaba Oláh, Csaba Váradi

**Affiliations:** 1Institute of Chemistry, Faculty of Materials Science and Chemical Engineering, University of Miskolc, 3515 Miskolc, Hungary; czamate@gmail.com (M.C.); bela.viskolcz@uni-miskolc.hu (B.V.); 2Department of Neurosurgery, Borsod-Abauj-Zemplen County Center Hospital and University Teaching Hospital, 3526 Miskolc, Hungary; olahcs@gmail.com; 3Institute of Energy, Ceramic and Polymer Technology, University of Miskolc, 3515 Miskolc, Hungary; attila.garami@uni-miskolc.hu; 4Department of Anesthesiology and Intensive Care, University of Pecs Medical School, 7624 Pecs, Hungary; csecsei.peter@pte.hu

**Keywords:** risk calculation, subarachnoid hemorrhage, cerebral vasospasm

## Abstract

**Background/Objectives:** Cerebral subarachnoid hemorrhage (SAH) from ruptured aneurysms poses significant morbidity and mortality risks. Among survivors, cerebral vasospasm can develop, increasing complications. This study investigates the relationship between blood parameters and the risk of vasospasm. **Methods:** We analyzed clinical data from patients with SAH—both with and without vasospasm—and healthy controls. Statistical analyses, including Spearman’s rank correlation and univariate analysis, were conducted. **Results:** Significant differences were observed between patients with and without vasospasm. Elevated white blood cell counts, a higher neutrophil-to-lymphocyte ratio, and lower platelet counts showed a significant association with symptomatic vasospasm. Younger age and female gender were associated with a higher risk. **Conclusions:** These preliminary findings highlight the importance of specific blood parameters and demographic factors in assessing the risk of cerebral vasospasm in SAH patients, supporting early risk stratification and monitoring to improve outcomes; however, these results require validation in larger cohorts.

## 1. Introduction

The rupture of an intracranial aneurysm resulting in subarachnoid hemorrhage (SAH) is associated with considerable morbidity and mortality [1]. Aneurysmal subarachnoid hemorrhage (aSAH) represents a prevalent cerebrovascular disorder, exhibiting a global incidence of approximately 7.9 per 100,000 individuals and a death rate nearing 40% [2]. The gravity of aSAH is compounded by the occurrence of cerebral vasospasm, a significant complication that affects between 40% and 70% of patients, with clinical manifestations evident in 20% to 40% of these cases [3]. Vasospasm typically emerges 3 to 21 days post-aSAH, leading to diminished cerebral blood flow and potentially triggering delayed cerebral ischemia (DCI) [4]. This hypoperfusion due to vasospasm and DCI is a critical factor contributing to the enduring neurological deficits and cognitive impairments observed in approximately 30% of individuals with aSAH who survive the initial aneurysm rupture [5].

Effective diagnosis and early intervention are paramount in managing aSAH and its complications [6]. The unpredictability of vasospasm presents a significant challenge in clinical settings, as its onset can occur after the initial hemorrhagic event, often when patients appear to be recovering [7]. The ability to predict vasospasm before clinical and radiological signs manifest remains an ongoing challenge [8]. Early diagnosis not only facilitates timely intervention but also improves patient outcomes by potentially reducing the incidence of DCI and associated neurological impairments [9].

Studies based on laboratory parameters and demographic data related to cerebral vasospasm following aSAH, including age, gender, platelet count, white blood cell counts, lymphocytes, neutrophils, hematocrit, and potassium levels, unveiled notable inconsistencies in their clinical implications [10]. Understanding these variables is critical, as they may serve as valuable biomarkers for identifying patients at increased risk of developing vasospasm.

Key findings from recent studies suggest a complex interplay between these factors and the risk of vasospasm [11]. Age-related findings indicate that younger patients, particularly those under 50 years, are at a significantly higher risk of developing vasospasm, while older patients may exhibit a reduced likelihood of this complication due to age-related vascular changes [3]. Gender differences also play a role; female patients tend to experience worse outcomes and are more susceptible to cerebral ischemia following aSAH [12]. Furthermore, hematological parameters such as elevated white blood cell counts (WBC), a high neutrophil-to-lymphocyte ratio (NLR), and low platelet counts have emerged as independent predictors of symptomatic vasospasm [13]. These findings underscore the necessity of incorporating a comprehensive assessment of these risk factors into clinical practice.

Given the critical role of these factors in influencing vasospasm risk, the aim of this study was to conduct an exploratory analysis on the correlation between quantitative blood parameters and other important variables to assess the risk of vasospasm development. By synthesizing findings from the literature and our analyses, we aim to enhance the understanding of these risk factors, ultimately improving management strategies and outcomes for patients experiencing aSAH. Through improved diagnostic accuracy and risk stratification, we can aim to mitigate the impact of cerebral vasospasm, reduce the burden of neurological deficits, and significantly enhance the quality of life for affected individuals.

## 2. Materials and Methods

Borsod County University Teaching Hospital is the largest hospital in Hungary. We treat an average of 60–70 patients per year for aneurysmal SAH. 90–95% of the procedures are performed using interventional radiology; the remaining procedures are performed using open neurosurgery procedures. In our previous prospective clinical study, we demonstrated that in the case of bleeding aneurysms, an altered pattern of serum N-glycome can appear.

### 2.1. Serum Samples

We collected serum samples from 22 healthy controls, 22 with SAH, and 22 with vasospasm; all of them were adults. Vasospasm was detected by TCD. Samples were collected at the Borsod Academic County Hospital (Miskolc, Hungary). Our retrospective analysis was conducted with research ethics approval. Regional Research Ethics Committee ethics approval number IG-102-102/2018.

The control group was considered healthy based on simple screening tests (abdominal ultrasound and chest X-ray), medical physical examination, medical history and laboratory parameters. Serum samples were collected on the 5th–6th day after the subarachnoid hemorrhage, and both patients with and without vasospasm were controlled with TCD, which is an accepted method for excluding and proving vasospasm. Cerebral vasospasm was defined radiologically as a mean flow velocity (MFV) > 120 cm/s in the Middle Cerebral Artery (MCA) via Transcranial Doppler (TCD). Patients with major comorbidities, such as a history of myocardial infarction, stroke, cancer, hematopoietic disease, were not included in the subarachnoid hemorrhage group; people were assessed based on their history and documentation before the ictus of aneurysmal rupture.

We have examined the following parameters: age, gender, modified Rankin scale (mRS) at admission, WFNS grade, serum sodium level, serum potassium level, serum creatinine level, serum carbamide level, C-reactive protein (CRP), serum white blood cell number (WBC), serum neutrophil number, serum lymphocyte number, hematocrit count (HTC), serum platelet count, and modified Rankin scale on the 90th day after admission.

We took several aspects into account when selecting the above-mentioned parameters. They were selected based on physiological, pathophysiological, neurological, and literature data. The WFNS score and the modified Rankin scale recorded at different times are internationally accepted for characterizing the neurological status of a patient with subarachnoid hemorrhage. We use general inflammatory parameters (CRP, WBC, serum neutrophil number, and serum lymphocyte number) to characterize the relationship between possible inflammatory changes in the body and the state of the vascular wall, because it has been proven that the mechanism of the vasospasm contains inflammatory processes. Blood rheological tests are essential for the examination of vasospasm, the components of which are hematocrit count (HTC) and serum platelet count. A characteristic of the physiological process is the altered cerebral vascular contractility with advancing age, which may influence the development of vasospasm. In the methods for examining vascular spasm, the examination of renal function values is also inevitable, which is why we selected serum sodium level, serum potassium level, serum creatinine level, and serum carbamide level.

### 2.2. Data Preprocessing and Normality Testing

The analysis began with a comprehensive preprocessing step, which included checking for missing values, standardizing clinical parameters, and removing extreme outliers. Normality testing was conducted using the Shapiro–Wilk test and D’Agostino’s K^2^ test to determine whether the data met parametric assumptions. Outliers were removed to ensure homogeneity for this pilot analysis, although we acknowledge this reduces the effective sample size.

### 2.3. Outlier Removal

To enhance the robustness of the analysis, extreme outliers were identified and removed. This step ensured that statistical tests were not unduly influenced by extreme values. A revised distribution of the clinical parameters was visually assessed using a raincloud plot.

### 2.4. Bivariate and Correlation Analysis

Spearman’s rank correlation analysis was performed to assess the relationships between key clinical variables. A correlation matrix and pair-plot analysis were generated to visually inspect potential nonlinear relationships and group-based differences between subarachnoid hemorrhage without vasospasm (SH) and subarachnoid hemorrhage with vasospasm (SH + V) patient groups.

### 2.5. Point Biserial Correlation Analysis

To evaluate associations between binary disease type (SH vs. SH + V) and continuous clinical variables, we conducted a Point Biserial correlation analysis. This statistical method allowed us to identify potential influencing factors affecting disease classification. Additionally, a threshold of *p* < 0.05 was used to determine statistical significance, and all correlations were examined in the context of potential clinical relevance.

### 2.6. Statistical Approach

Given the mix of normally and non-normally distributed variables, non-parametric statistical methods were employed. Mann–Whitney U tests and Kruskal–Wallis tests were used for group comparisons, while Spearman’s correlation was applied to assess associations between continuous variables. Logistic regression and potential multivariate analyses were considered to further explore significant findings.

## 3. Results

### 3.1. Univariate Analysis and Normality Testing

As an initial step in the exploratory data analysis, we conducted a univariate analysis to assess the distribution of key variables. Normality was tested using both the Shapiro–Wilk test and D’Agostino’s K^2^ test, with results summarized in Table 1. Additionally, a raincloud plot was used to visually inspect the data distribution.

### 3.2. Normality Assessment

Based on the statistical tests, the following observations were made:Normally distributed variables: Variables such as age, Na (sodium), K (potassium), HTC (hematocrit), and platelet count did not show significant deviations from normality (*p* > 0.05 in both tests).Non-normally distributed variables: Several variables, including baseline mRS, WFNS grade, creatinine, carbamide, CRP, WBC, neutrophil, lymphocyte, and discharge mRS at 90 days, exhibited significant deviations from normality (*p* < 0.05 in at least one test).

These findings suggest that many key clinical variables do not follow a normal distribution, necessitating the use of non-parametric statistical methods in further analyses.

### 3.3. Visual Inspection

The raincloud plot (Figure 1) further supports the normality test results by illustrating the data distribution for each variable. Symmetrical, bell-shaped distributions were observed for the normally distributed variables, while skewed distributions were evident for non-normally distributed ones. Notably, baseline mRS, WFNS grade, and inflammatory markers (CRP, WBC, neutrophils, lymphocytes) showed strong deviations from normality with notable skewness and outliers. The raincloud plot also proved useful for identifying potential outliers.

### 3.4. Outlier Removal and Reassessment

To improve data quality and ensure robust statistical analysis, we identified and removed outliers from the dataset. A revised raincloud plot (Figure 2) was generated to visualize the distribution after outlier removal. The updated plot indicates a reduction in extreme values while preserving the overall distribution trends. Outliers were identified for each variable using the interquartile range (IQR) method; values below Q1 − 1.5 × IQR or above Q3 + 1.5 × IQR were considered extreme. To maintain clinical coherence of laboratory panels and to ensure consistent sample sizes across multivariate analyses and visualizations (e.g., raincloud plots, pair-plots, and correlation heatmaps), a case-wise (listwise) deletion strategy was applied: if a record contained an outlier in any variable, the entire case was excluded from further analyses. This conservative approach reduces the influence of single extreme values and improves comparability across models and figures. The number of excluded records per variable, as well as pre- and post-filtering sample sizes, are reported in Appendix A.

After removing outliers, we re-evaluated normality using the Shapiro–Wilk and D’Agostino’s K^2^ tests. While some variables showed improved adherence to normality assumptions, others remained non-normally distributed. This further supports the need for nonparametric statistical methods in subsequent analyses.

### 3.5. Bivariate Analysis: Spearman Correlation and Pairwise Relationships

To assess relationships between variables, we computed Spearman’s rank correlation coefficients and visualized the results using a heatmap (Figure 3). The correlation matrix provides insights into the strength and direction of associations between clinical parameters. Neurological severity (WFNS grade and baseline mRS) is a strong indicator of outcome and likely vasospasm risk. Systemic inflammation (CRP, WBC, neutrophils) appears linked to worse neurological status, which could contribute to vasospasm development or severity. Monitoring inflammatory markers, alongside neurological assessments, might help manage vasospasm risk in patients. Kidney function (creatinine, carbamide) and electrolytes (Na, K) show weaker or no clear direct correlation with vasospasm severity or outcomes.

Additionally, a pair-plot analysis (Figure 4) was performed to explore potential nonlinear patterns and interactions between variables. The pair-plot distinguishes between the two patient groups (SH vs. SH + V) using color coding, allowing for visual assessment of possible group-based differences. Clinically, these findings emphasize the importance of considering multiple factors when assessing patient outcomes. The lack of strong, obvious correlations between variables may indicate the complexity of the patient profiles and the multifactorial nature of the disease or condition under study.

### 3.6. Binary Logistic Regression Results

Table 2 summarizes the key factors for cerebral vasospasm as identified in the study’s analysis of 44 patients (22 vasospasm vs. 22 non-vasospasm). Clinical interventions at the hospital primarily utilized interventional radiology (38 coiling cases), with the remaining 6 cases treated via open neurosurgery (clipping).

In summary, the exploratory data analysis identified significant deviations from normality in several key clinical variables, particularly those related to baseline neurological status (e.g., mRS, WFNS grade) and inflammatory markers (e.g., CRP, WBC, neutrophils, lymphocytes). Normality testing using the Shapiro–Wilk and D’Agostino’s K^2^ tests, supported by raincloud plot visualizations, demonstrated that while some variables, such as age, sodium, potassium, hematocrit, and platelet count, were normally distributed, many others were not. These findings highlight the need to apply nonparametric statistical methods for accurate analysis and interpretation.

Outliers were systematically identified and removed using the interquartile range (IQR) method. To maintain the integrity of multivariate analyses and visual consistency across visualizations, a conservative case-wise deletion strategy was employed—excluding entire records if any variable within them was considered an outlier. This approach enhanced data quality, minimized bias from extreme values, and ensured consistent sample sizes across analyses. Post-filtering normality was reassessed; although some improvement was observed, several variables remained non-normally distributed, further justifying the use of non-parametric techniques.

Finally, bivariate analysis using Spearman’s rank correlation and pair-plot visualizations allowed for the exploration of associations between variables and potential differences between patient subgroups (SH vs. SH + V). These analyses provided deeper insights into underlying relationships, setting the stage for robust multivariate modeling in the subsequent phases of the study.

## 4. Discussion

Our study provides valuable insights into the relationship between clinical factors and the risk of cerebral vasospasm following aneurysmal subarachnoid hemorrhage (aSAH). We identified elevated white blood cell counts, a higher neutrophil-to-lymphocyte ratio (NLR), and lower platelet counts as significant predictors of symptomatic vasospasm, alongside younger age and female gender. The elevated risk of vasospasm in younger patients aligns with findings from Magge et al., who reported that individuals under 50 years are more susceptible to this complication [14]. Our results contribute to this body of literature by emphasizing the role of inflammatory markers, particularly NLR, as critical predictors in this age group. This is consistent with the broader understanding that systemic inflammation plays a pivotal role in the pathophysiology of vasospasm. Dodd et al. highlighted that inflammatory responses following SAH can exacerbate vascular injury, leading to vasospasm development [4]. Our findings reinforce the importance of monitoring inflammatory markers in clinical settings. Additionally, our observation that female patients are at a higher risk of symptomatic vasospasm supports previous research, including studies by Han et al., which noted gender disparities in outcomes after aSAH [15]. Understanding these gender differences is vital for developing tailored management strategies that address the unique risks faced by female patients. Methodologically, we employed a multivariate logistic regression model to control for confounding variables such as initial bleed severity (Fisher Scale) and treatment modality (surgical clipping versus endovascular coiling). This rigorous approach enhances the reliability of our findings and addresses limitations present in studies with smaller sample sizes or insufficient control for confounders, as noted by Pavelka et al. [3]. Notably, even after adjusting for these significant factors, NLR and age remained independent predictors of vasospasm, underscoring their clinical relevance.

Our study has several limitations that should be noted. First, the small sample size (*n* = 22 per group) limits statistical power and precludes robust multivariable regression analysis. Second, the retrospective design and the case-wise deletion of outliers may have introduced selection bias. Additionally, vasospasm was defined based on TCD velocities without angiographic confirmation in all cases. Consequently, our results should be interpreted as hypothesis-generating and require validation in larger, multi-center studies. Despite these limitations, we highlighted the utility of bedside markers. While the Fisher and Hunt & Hess scales provide “snapshots” at admission, inflammatory markers like the Neutrophil-to-Lymphocyte Ratio (NLR) offer dynamic, real-time data that can be monitored daily to adjust clinical suspicion of impending vasospasm. NLR and platelet counts are “low-cost, rapidly available” tools that provide insights beyond what the initial CT scan (Fisher scale) conveys.

## 5. Conclusions

In conclusion, our findings underscore the necessity of a comprehensive approach to risk assessment in patients with aSAH. By integrating demographic factors, inflammatory markers, and hematological parameters into clinical decision-making, healthcare providers can enhance their ability to identify patients at risk for cerebral vasospasm and implement targeted monitoring and treatment strategies. Regarding the clinical signs, while DCI is the ultimate clinical concern, radiological vasospasm remains the primary physiological driver and an essential early warning sign in the neuro-ICU setting. Future research should focus on longitudinal studies that further explore the mechanisms underlying these relationships, along with randomized controlled trials to evaluate the efficacy of tailored interventions based on identified risk factors. Ultimately, improving our understanding of the predictors of vasospasm will contribute to better outcomes for patients affected by this serious complication of aSAH.

## Figures and Tables

**Figure 1 life-16-00059-f001:**
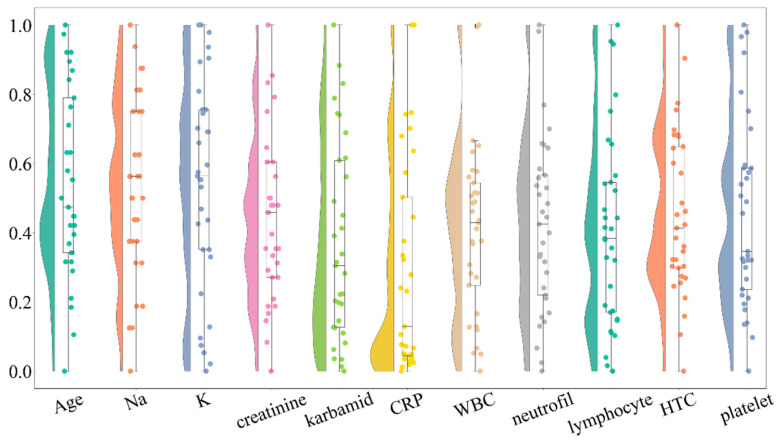
Raincloud plot illustrating the distribution of key clinical variables in the study population. The plot combines violin plots, which show the distribution shape, with boxplots indicating median and interquartile ranges, along with individual data points represented as scatter points. Variables include age, baseline modified Rankin scale (mRS), WFNS grade, serum sodium (Na), potassium (K), C-reactive protein (CRP), white blood cell count (WBC), neutrophil count, lymphocyte count, hematocrit count (HTC), and platelet count. The distributions reveal significant deviations from normality for several critical clinical parameters, particularly those related to inflammatory markers and neurological status.

**Figure 2 life-16-00059-f002:**
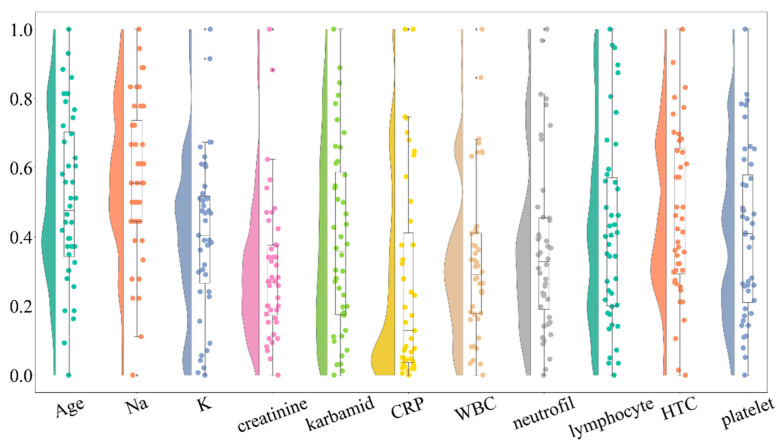
Raincloud plot displaying the distribution of clinical variables following the removal of extreme outliers. This plot illustrates the improvement in the distribution of variables while maintaining the overall trends observed in the data. Outliers were identified using the interquartile range (IQR) method, and their removal enhanced the robustness of subsequent statistical analyses. The revised distributions are crucial for ensuring accurate statistical interpretations in the context of cerebral vasospasm risk.

**Figure 3 life-16-00059-f003:**
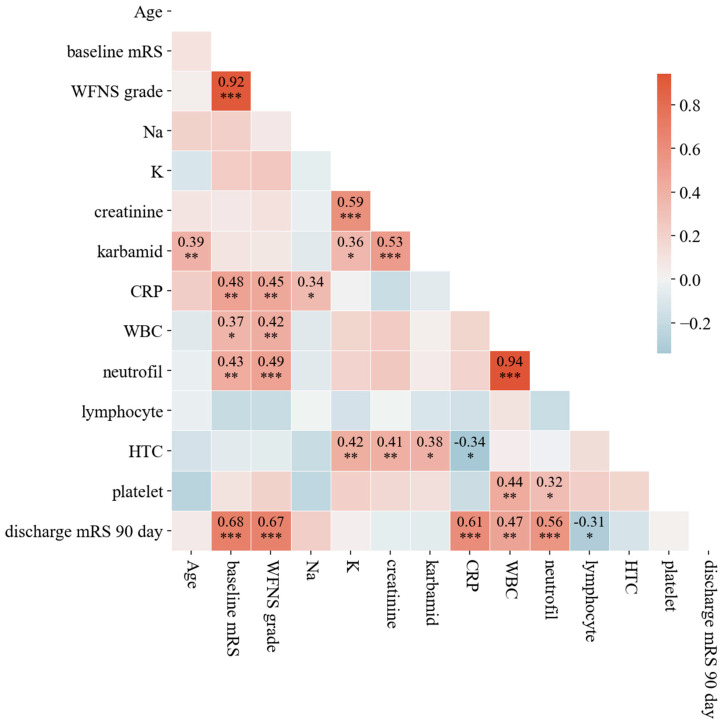
Spearman’s rank correlation coefficients among key clinical parameters assessed in the study. The matrix highlights the strength and direction of associations between variables. Notably, strong correlations are observed between neurological severity indicators (WFNS grade and baseline mRS) and inflammatory markers (CRP, WBC, and neutrophils), suggesting that elevated inflammatory responses may contribute to worse neurological outcomes and an increased risk of symptomatic vasospasm. The color gradient indicates positive and negative correlations, with darker shades representing stronger associations. Stars are used to indicate statistical significance at three commonly utilized levels. A *p*-value less than 0.05 is marked with (∗), less than 0.01 with (∗∗), and less than 0.001 with (∗∗∗).

**Figure 4 life-16-00059-f004:**
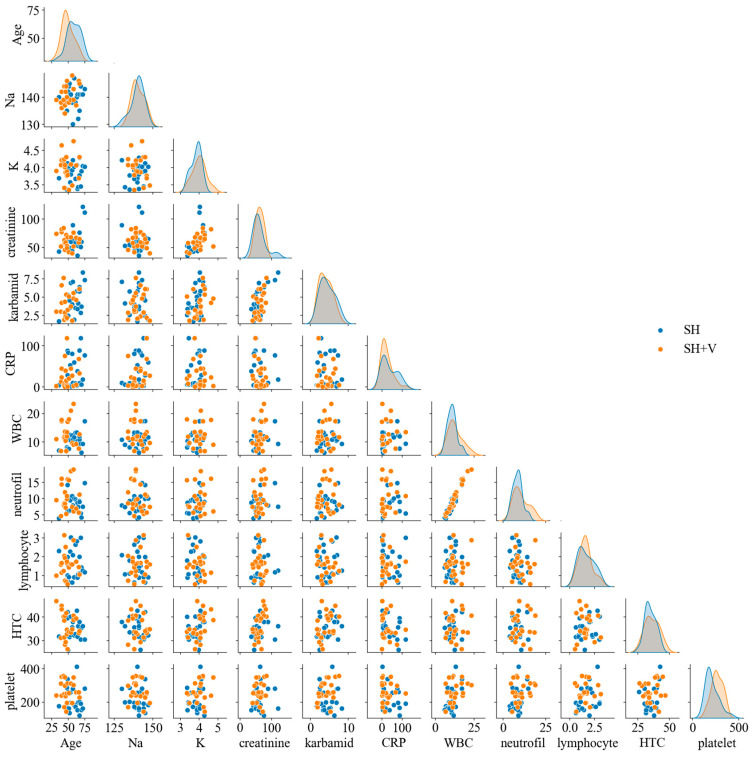
Pair-plots distinguishing between the two patient groups: those with subarachnoid hemorrhage without vasospasm (SH) and those with subarachnoid hemorrhage with vasospasm (SH + V). Each subplot depicts the relationships between pairs of clinical variables, utilizing color coding to differentiate between the two groups. The pair-plot allows for visual assessment of potential nonlinear patterns and interactions, revealing significant differences in the distribution of variables between the two groups. This analysis emphasizes the complexity of patient profiles and the multifactorial nature of the risk for cerebral vasospasm.

**Table 1 life-16-00059-t001:** Summary of the results of normality testing for key clinical variables assessed in the study population. The table presents the sample size for each variable, along with the *p*-values from the Shapiro–Wilk test and D’Agostino’s K^2^ test.

Name	Sample Size	Shapiro–Wilk Test	*p*-Value (Shapiro)	D’Agostino’s K^2^ Test	*p*-Value (D’Ago.)
Age	44	1	0.68	1	0.45
baseline mRS	44	0	0.00	0	0.00
WFNS grade	44	0	0.00	0	0.00
Na	44	1	0.77	1	0.67
K	43	1	0.21	1	0.79
creatinine	44	0	0.00	0	0.00
karbamid	44	1	0.09	1	0.16
CRP	43	0	0.00	0	0.02
WBC	44	0	0.00	0	0.01
neutrofil	44	0	0.01	0	0.05
lymphocyte	43	1	0.06	1	0.23
HTC	44	1	0.49	1	0.54
platelet	44	1	0.20	1	0.32
discharge mRS 90 day	44	0	0.00	0	0.04

**Table 2 life-16-00059-t002:** The final model identifies elevated WBCs, a higher NLR, and lower platelet counts as the strongest independent predictors for symptomatic vasospasm in aSAH patients. Younger patients and female patients were also found to be at a significantly higher risk.

Independent Variable	Statistical Significance (*p*-Value)	Clinical Finding
NLR (Neutrophil-to-Lymphocyte Ratio)	*p* < 0.05	Significant Independent Factor. A high NLR is strongly associated with an increased risk of symptomatic vasospasm.
WBC (White Blood Cell Count)	*p* < 0.05	Significant Independent Factor. Elevated white blood cell counts are indicative of the systemic inflammatory response linked to vasospasm development.
Age	Significant	Negative Factor. Younger patients (especially those under 50) have a significantly higher risk compared to older patients.
Therapy (Coiling vs. Clipping)	Descriptive Only	Intervention Profile. 38 patients (90–95%) underwent coiling, while 6 patients (5–10%) underwent clipping.
WFNS Grade/baseline mRS	Significant Correlation	Severity Indicator. Strong correlations exist between high neurological severity grades and elevated inflammatory markers (CRP, WBC).
Hunt and Hess/Fisher Scale	High Correlation	Often used interchangeably with WFNS to characterize the patient’s baseline neurological status and risk level.
Platelet Count	Significant	Independent Factor. Low platelet counts are associated with a higher likelihood of vasospasm.

## Data Availability

The data presented in this study are available on request from the corresponding author. The data are not publicly available due to privacy or ethical restrictions.

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
