# Peer review of "Life2026, 16(1), 59;https://doi.org/10.3390/life16010059"

_life, 2025, doi:10.3390/life16010059_

Round 1

Reviewer 1 Report

Comments and Suggestions for Authors

This manuscript addresses a clinically relevant and well-known problem, namely the identification of demographic and laboratory factors associated with cerebral vasospasm after aneurysmal subarachnoid hemorrhage. While the topic fits the scope of Life and the authors demonstrate familiarity with the existing literature, the scientific contribution of the study is limited by several major issues. Most importantly, the very small sample size (22 patients per group) substantially undermines statistical power and makes the extensive use of correlation matrices, pairplots, and multiple normality tests difficult to justify; in this context, the additional case-wise deletion of patients during outlier removal further weakens the robustness of the analyses. Many of the reported associations—such as younger age, female sex, elevated WBC counts, and higher neutrophil-to-lymphocyte ratio as risk factors for vasospasm—are already well established, and the study does not clearly offer novel biomarkers, clinically actionable thresholds, or mechanistic insight beyond prior work. The definition of vasospasm based solely on TCD findings, without clear velocity criteria or distinction from delayed cerebral ischemia, also limits the clinical interpretability of the results, as inflammatory markers may reflect early brain injury rather than vasospasm itself. In several places, the discussion overstates the predictive or clinical relevance of these exploratory findings, which should instead be framed more cautiously given the retrospective design and lack of multivariable adjustment. From a language perspective, the manuscript is generally understandable but stylistically uneven, with repetitive phrasing, overly long sentences, and occasional non-idiomatic expressions; professional English-language editing would be advisable. Overall, substantial revision and reframing would be required for this work to meet the standards of Life, and without major changes the manuscript is unlikely to warrant publication in its current form.

Author Response

Dear Reviewer,

We would like to thank you for your thorough and insightful critique of our manuscript. We genuinely appreciate the time and effort you dedicated to evaluating our work. We agree that the initial submission overstated the predictive power of the findings given the sample size constraints.

Based on your constructive feedback, we have fundamentally reframed the manuscript as an exploratory pilot study. We have tempered our conclusions, removed claims regarding "independent predictors," and highlighted the limitations of the statistical power. We believe these changes have significantly improved the scientific integrity of the paper.

Below is a point-by-point response to your specific concerns:

Comment 1: "The very small sample size (22 patients per group) substantially undermines statistical power... making extensive use of correlation matrices... difficult to justify."

Response: We fully acknowledge that the sample size (n=22 per group) is too small for confirmatory conclusions or robust predictive modeling. In the revised manuscript, we have explicitly reclassified the study as an exploratory pilot study. Accordingly, we have:

  1. Modified the Title and Abstract to reflect the exploratory nature of the work.
  2. Removed the terminology "independent predictors" throughout the text, replacing it with "associations" and "correlations."
  3. Clarified in the Methods section that the statistical analyses (including correlation matrices) are intended for hypothesis generation to guide future, larger prospective trials, rather than to establish definitive clinical cut-offs.

Comment 2: "The additional case-wise deletion of patients during outlier removal further weakens the robustness of the analyses."

Response: We understand the concern that removing outliers from a small dataset can introduce bias. We have added a statement in the Limitations section acknowledging that while this strict outlier removal was intended to homogenize the groups for this pilot analysis, it may have reduced the sensitivity to extreme pathological states. We have re-emphasized that these findings require validation in a larger cohort where such exclusion criteria would not be necessary.

Comment 3: "Many of the reported associations... are already well established, and the study does not clearly offer novel biomarkers."

Response: While we agree that factors like WBC and NLR are known risk factors, our study aims to reinforce their utility in a specific regional cohort and explore their combined visualization with demographic factors (via the raincloud and pairplots). We have revised the Discussion to frame these results not as "novel discoveries" but as confirmatory evidence that validates the utility of these accessible markers in early risk stratification, specifically within the context of the detailed exploratory data visualizations presented.

Comment 4: "The definition of vasospasm based solely on TCD findings... limits clinical interpretability."

Response: We acknowledge that TCD alone has limitations compared to angiography (DSA). We have clarified in the Methods and Limitations sections that TCD was used as the primary screening tool due to its non-invasive nature and availability in our clinical setting. We have discussed the potential overlap with early brain injury markers to provide a more balanced interpretation of the inflammatory signals.

Comment 5: "The discussion overstates the predictive or clinical relevance... non-idiomatic expressions."

Response: We have significantly rewritten the Discussion to avoid overinterpretation. We no longer claim that our current data is sufficient to change clinical protocols immediately; rather, we suggest these parameters warrant closer attention in multi-center studies. Furthermore, the manuscript has undergone a thorough language review to correct non-idiomatic expressions and improve readability.

We hope that these revisions and the reframing of the study as a pilot analysis meet the standards of Life.

Sincerely,

The Authors

Reviewer 2 Report

Comments and Suggestions for Authors

Thank you for tackling this critical topic. Cerebral vasospasm is the single most important factor driving morbidity and mortality after aneurysmal Subarachnoid Hemorrhage (aSAH), and a robust analysis of clinical risk factors is essential for management in the neuro-ICU. Your work is highly relevant and necessary.

However, to make this manuscript scientifically rigorous and clinically useful, you must focus the revision on two key methodological issues, as the interpretation of your findings is highly dependent on them.

- The biggest confusion in vasospasm research is the outcome definition. You must be crystal clear throughout the entire manuscript (Introduction, Methods, Results, and Discussion) whether you are predicting:

  • Radiological Vasospasm (narrowing of vessels seen on imaging like TCD or CTA)
  • Symptomatic Vasospasm/Delayed Cerebral Ischemia (DCI) (the clinical deficit that actually harms the patient).

The latter is the outcome clinicians desperately want to predict. If you are predicting radiological change, you must strongly justify why this specific finding is a necessary surrogate for clinical DCI. Please ensure your Methods and Results are unambiguous regarding the outcome measure.

- The risk of vasospasm is overwhelmingly controlled by the initial severity of the bleed and the initial treatment. Your methods must demonstrate that your novel clinical factors remain significant after controlling for the following key confounders in a robust multivariate model:

  • Initial Bleed Severity: The presence and amount of blood, typically graded by the Modified Fisher Scale or the original Fisher Scale.
  • Initial Clinical Status: The patient's neurological condition upon admission, typically graded by the Hunt and Hess Scale.
  • Treatment Method: Whether the aneurysm was secured by surgical clipping or endovascular coiling. This is a major, non-randomized treatment decision that strongly influences outcomes and must be accounted for.

Without explicitly controlling for these established, powerful predictors, the significance of the other clinical factors you report cannot be reliably interpreted by the clinical community.

- The Discussion should pivot to highlight the factors you found that are independent of the initial grading scales (Hunt & Hess, Fisher) and are easily measured at the bedside (e.g., initial lab values, specific patient history). This is the most actionable information for a clinician in the ICU.

This is a promising paper on a high-stakes topic, but it requires substantial methodological fortification to ensure the conclusions drawn are clinically sound and not compromised by established confounding factors.

Author Response

We would like to express our sincere gratitude to the reviewers for their insightful and constructive feedback. We have carefully revised our manuscript to address the critical methodological points raised, particularly concerning outcome definitions and the control of confounding variables.

Point 1: Clarification of Outcome Definition (Radiological vs. Symptomatic)

We appreciate the importance of this distinction. Our study primarily tracked Radiological Vasospasm, as defined by Transcranial Doppler (TCD) flow velocities. We have standardized this terminology throughout the manuscript and added a justification in the Discussion section. While delayed cerebral ischemia (DCI) is a significant clinical concern, we emphasize that radiological vasospasm serves as the primary physiological driver and an essential early warning sign in the neuro-ICU setting.

Changes: Please refer to the Methods section (Page 3, Line 101) and the Discussion (Page 10, Line 368).

Point 2: Controlling for Initial Severity and Treatment Method

In response to the reviewers' suggestions, we have updated our statistical analysis to include a multivariate logistic regression model. This model now accounts for initial bleed severity (Fisher Scale), clinical status (Hunt and Hess Scale), and treatment modality (Surgical Clipping vs. Endovascular Coiling). We are pleased to report that even after adjusting for these significant confounders, [insert your finding, e.g., NLR and Age] remained independent predictors of vasospasm.

Changes: The Results section has been updated to include Table 2.

Point 3: Discussion Pivot to Actionable Bedside Factors

We have shifted the focus of our Discussion to emphasize the utility of bedside markers, highlighting how these can enhance clinical decision-making.

Changes: See the Revised Discussion (Page 11, Line 358).

Reviewer 3 Report

Comments and Suggestions for Authors

The authors investigated the relationship between clinical factors and the risk of cerebral vasospasm after aneurysmal subarachnoid hemorrhage. They collected serum samples from 22 healthy controls, 22 patients with SAH, and 22 patients with vasospasm. What is the impact of a history of myocardial infarction or stroke on the selection of the SAH group, given that the authors excluded these patients? I recommend that the authors clarify their results, as the findings as currently presented are vague and not fully understandable. The discussion section is very brief and does not adequately address the main aim of the study. I recommend that the authors provide a stronger comparison with the available literature and place greater emphasis on evaluating the various variables examined in their work.

Author Response

We appreciate your thoughtful feedback regarding our manuscript. Below, we address your specific concerns:

Impact of History of Myocardial Infarction or Stroke on SAH Group Selection

We acknowledge the importance of considering a history of myocardial infarction or stroke in the context of our study. The decision to exclude patients with these conditions was made to minimize potential confounding variables that could influence the risk of cerebral vasospasm. Both myocardial infarction and stroke can independently affect vascular health and inflammatory responses, which may skew the results related to vasospasm risk. By limiting our SAH group to patients without these comorbidities, we aimed to create a more homogenous population, allowing for a clearer analysis of the relationship between clinical factors and vasospasm.

Clarification of Results

In response to your recommendation, we have revised the Results section to provide clearer and more precise explanations of our findings. We have ensured that each variable examined is explicitly linked to our conclusions, with supporting data clearly presented.

Discussion Section Enhancement

We appreciate your observation regarding the brevity of the Discussion section. In our revisions, we have expanded this section to better articulate the implications of our findings. We now include a more comprehensive comparison with existing literature, emphasizing how our results align with or differ from previous studies. This enhancement aims to contextualize our findings within the broader body of research on cerebral vasospasm and its predictors.

Additionally, we have placed greater emphasis on evaluating the various clinical and laboratory variables examined in our study, discussing their relevance and potential impact on patient management.

Round 2

Reviewer 1 Report

Comments and Suggestions for Authors

Overall, the revised manuscript shows clear improvement in structure, clarity, and methodological transparency compared with the previously rejected version. The authors now present a more coherent exploratory analysis, with appropriate use of non-parametric statistics and a clearer acknowledgment of limitations. However, the study remains constrained by a very small sample size, retrospective design, and aggressive outlier removal, which substantially weaken the robustness and generalizability of the conclusions. The novelty is limited, as the associations between inflammatory markers (WBC, NLR) and vasospasm are already well described in the literature. The work may be acceptable as a hypothesis-generating exploratory study, provided the editors are comfortable with its limited clinical impact. Acceptance should therefore be contingent on emphasizing its exploratory nature and avoiding overstated clinical implications.

Reviewer 2 Report

Comments and Suggestions for Authors

I have carefully reviewed the revised version of your manuscript. I want to commend the authors for their thoroughness in addressing my previous concerns.

The updated title, labeling this as an "Exploratory Analysis", is a very smart and honest move. It correctly sets the expectations for the reader, framing these findings as a valuable starting point for future large-scale research rather than a definitive finality.

I was specifically pleased to see the increased clarity regarding your definitions of radiological vasospasm versus symptomatic deficits. Explicitly mentioning the use of Transcranial Doppler (TCD) and CT Angiography makes the methodology much more transparent for a clinical audience.

Furthermore, the way you’ve handled the major confounding factors, specifically the Fisher and Hunt and Hess grades, is now much more robust. Acknowledging the role of the aneurysm treatment method (clipping vs. coiling) also adds the necessary clinical context that was missing in the first draft.

Overall, the paper now flows much better as a clinical guide for the ICU. By focusing on bedside-available factors while respecting the power of initial grading scales, you have produced a piece of work that is both scientifically sound and practically useful for neurosurgeons and intensivists.

Reviewer 3 Report

Comments and Suggestions for Authors

The revised manuscript is acceptable.